# High resolution cryo-EM structure of the helical RNA-bound Hantaan virus nucleocapsid reveals its assembly mechanisms

Benoît Arragain[1], Juan Reguera[2], Ambroise Desfosses[1], Irina Gutsche[1], Guy Schoehn[1]*, Hélène Malet[1]*

[1]Electron Microscopy and Methods Group, Université Grenoble Alpes, CNRS, CEA, Institute for Structural Biology, Grenoble, France; [2]Complexes Macromoléculaires Viraux, Aix-Marseille Université, CNRS, INSERM, AFMB UMR 7257, Marseille, France

**Abstract** Negative-strand RNA viruses condense their genome into helical nucleocapsids that constitute essential templates for viral replication and transcription. The intrinsic flexibility of nucleocapsids usually prevents their full-length structural characterisation at high resolution. Here, we describe purification of full-length recombinant metastable helical nucleocapsid of Hantaan virus (*Hantaviridae* family, *Bunyavirales* order) and determine its structure at 3.3 Å resolution by cryo-electron microscopy. The structure reveals the mechanisms of helical multimerisation via sub-domain exchanges between protomers and highlights nucleotide positions in a continuous positively charged groove compatible with viral genome binding. It uncovers key sites for future structure-based design of antivirals that are currently lacking to counteract life-threatening hantavirus infections. The structure also suggests a model of nucleoprotein-polymerase interaction that would enable replication and transcription solely upon local disruption of the nucleocapsid.
DOI: https://doi.org/10.7554/eLife.43075.001

*For correspondence:
guy.schoehn@ibs.fr (GS);
helene.malet@ibs.fr (HM)

**Competing interests:** The authors declare that no competing interests exist.

## Introduction

The *Bunyavirales* order is one of the largest groups of segmented negative-strand RNA viruses (sNSV) that include many pathogenic strains (*Sun et al., 2018*). In particular, the *Hantaviridae* family comprises the virus Hantaan (HTNV) that gives rise to haemorrhagic fevers with renal syndrome and the virus Sin Nombre that is linked to severe pulmonary illnesses with fatality rates up to 40%. Neither treatment nor vaccine is currently available to counteract them.

The *Bunyavirales* genome is usually divided into three RNA segments enwrapped by the viral nucleoproteins (NP). The resulting nucleocapsids (NCs) protect the genome and serve as a replication/transcription template for the viral polymerase (*Reuter and Krüger, 2018*). They coat the genomic and anti-genomic RNA during replication but not the mRNA produced by transcription. As they are specific and essential to the viral cycle, NCs constitute an attractive potential target for antiviral drugs.

Nucleoproteins of segmented NSV (sNSV) present a large diversity of folds (*Sun et al., 2018*). Most of the available structures have been determined as rings and monomers that present the advantage of being rigid enough for crystallisation. However, the relevant conformations of assembled NPs in the viral context correspond to flexible helices or pearl-necklaces that encapsulate long RNA segments. Helical crystal structures of La Crosse virus NP (*Peribunyaviridae*, LACV) (*Reguera et al., 2013*) and Crimean Congo Fever Virus NP (*Nairoviridae*, CCFHV) (*Wang et al.,*

**eLife digest** Rats and mice sometimes transmit hantaviruses, a family of microbes that can cause deadly human diseases. For example, the Hantaan virus leads to haemorrhagic fevers that are potentially fatal. There are no vaccine or even drugs against these infections.

To multiply, viruses must insert their genetic material inside a cell. While the body often detects and destroys foreign genetic information, hantaviruses can still evade our defences. Molecules called nucleoproteins bind to the viral genome, hiding it away in long helices called nucleocapsids. When the virus needs to replicate, an enzyme opens up the nucleocapsid, reads and copies the genetic code, and then closes the helix. Yet, researchers know little about the details of this process, or even the structure of the nucleocapsid.

Here, Arragain et al. use a method called cryo-electron microscopy to examine and piece together the exact 3D structure of the Hantaan virus nucleocapsid. This was possible because the new technique allows scientists to observe biological molecules at an unprecedented, near atomic resolution. The resulting model reveals that the viral genome nests into a groove inside the nucleocapsid. It also shows that specific interactions between nucleoproteins stabilise the helix. Finally, the model helps to provide hypotheses on how the enzyme could read the genome without breaking the capsid.

Mapping out the structure and the interactions of the nucleocapsid is the first step towards finding molecules that could destabilise the helix and neutralise the virus: this could help fight both the Hantaan virus and other members of its deadly family.

DOI: https://doi.org/10.7554/eLife.43075.002

*2012*) have been determined but they correspond to local organisations of pearl-necklace-like native NCs. Influenza double-helical NCs 3D structures have been described at low resolution but display different helical parameters, also reflecting their malleability (*Arranz et al., 2012*; *Moeller et al., 2012*). NC flexibility appears as a hallmark in NSVs as NCs from non-segmented NSV such as Ebola or measles virus require C-terminal NP truncations in order to obtain rigid NCs and high-resolution 3D structures (*Gutsche et al., 2015*; *Sugita et al., 2018*; *Wan et al., 2017*). In this context, HTNV-NCs are intriguing as they were shown to be able to form rather rigid helices of 10 nm diameter within viruses (*Battisti et al., 2011*; *Huiskonen et al., 2010*) and during cell infection (*Goldsmith et al., 1995*). We therefore aimed at obtaining their high-resolution 3D structure, identifying the determinants of NP polymerisation and visualising RNA organisation.

## Results

Expression of recombinant full-length HTNV-NP in insect cells led to formation of recombinant NCs that have a diameter consistent with native NCs (*Figure 1—figure supplement 1*). Cryo-EM images collected on a Titan Krios enabled structural determination of HTNV-NC at 3.3 Å resolution (*Figure 1A*, *Video 1*, *Figure 1—figure supplements 1* and *2*, *Figure 1—source data 1*). HTNV-NC is a left-handed helix with a pitch of 68.03 Å and 3.6 subunits per turn (*Figure 1A* and *Figure 1—figure supplement 1*). To derive an atomic model of HTNV-NP, the monomeric crystal structure of HTNV-NP comprising residues 113 to 429 (*Olal and Daumke, 2016*) was fitted into the EM map, and the N-terminal residues and loops not present in the crystal structures were unambiguously built (*Figure 1D*). The HTNV-NC model was then iteratively rebuilt and the all-atom model refined using stereochemical restraints.

Although we expressed the full-length NP, residues 1 to 79 are missing in the final map, indicating that they do not follow the helical symmetry. Interestingly, a $NP_{74-429}$ construct obtained by trypsin limited proteolysis still forms a rigid helix, which means that the $N-ter_{1-73}$ is not necessary for helix stabilisation (*Figure 2—figure supplement 1A,B*). The $N-ter_{1-73}$ might correspond to a flexibly-linked coiled-coil as previously visualised in the structure of an N-ter construct (*Boudko et al., 2007*).

The structure of $NP_{80-429}$ is composed of a core comprising residues 117 to 398 ($NP_{core}$) that defines two lobes surrounding a positively charged groove (*Figure 1B,C*, *Video 1*). The $NP_{core}$ structure is relatively conserved compared to the apo monomeric truncated crystal structure, with a Cα

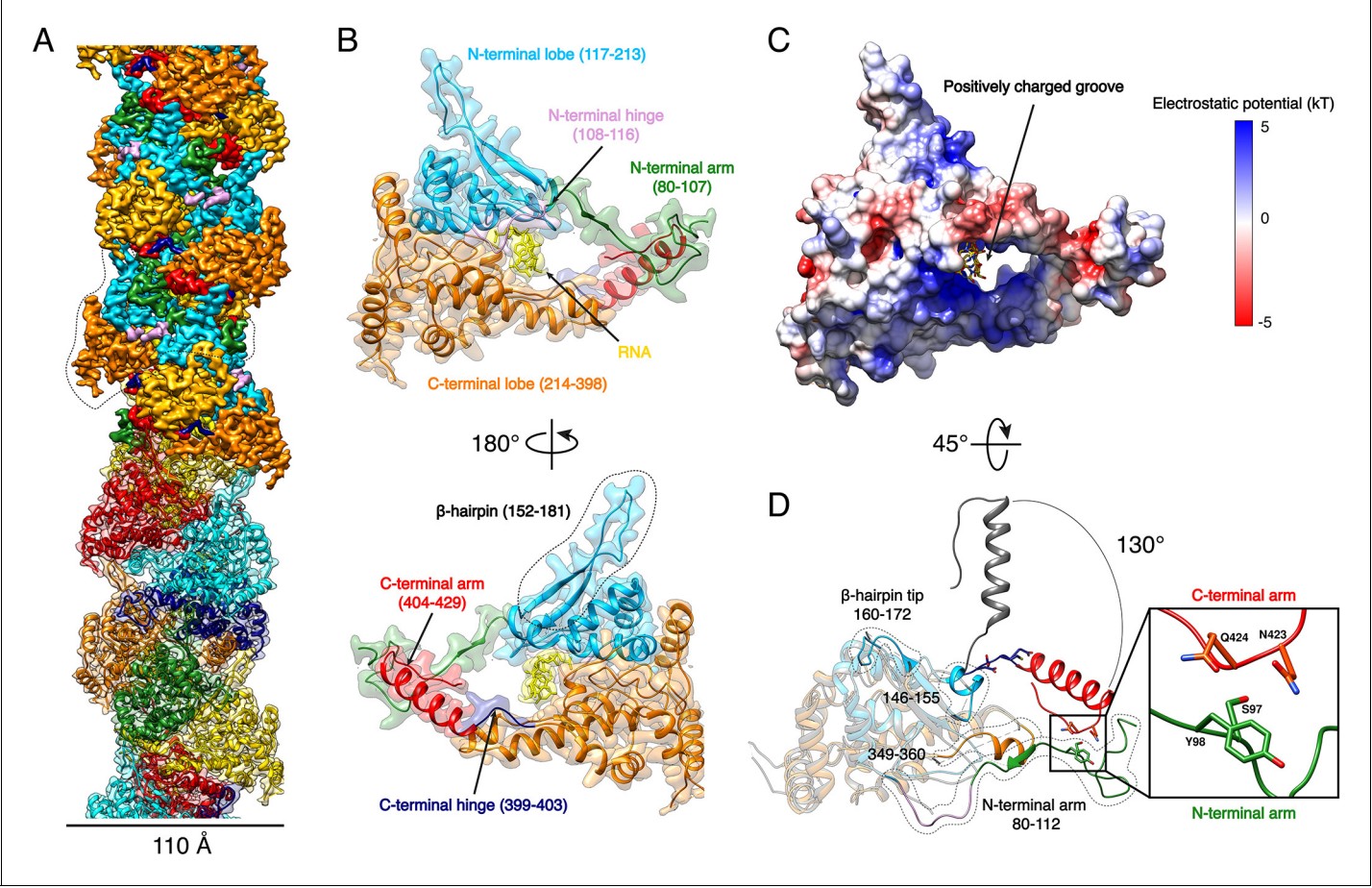

**Figure 1.** HTNV-NC structure. (**A**) HTNV-NC helical organisation. The upper part of the cryo-EM map is coloured per NP domain, whereas the lower part is shown as transparent and coloured per NP protomer. The NP surrounded by dotted lines corresponds to *Figure 1B* orientation. (**B**) Domain organisation of one monomer extracted from the NC helical assembly. Each domain is coloured as in *Figure 1A* upper part. The newly described β-hairpin (residues 152–181) is labelled and surrounded by dotted lines. (**C**) Electrostatic surface representation of HTNV-NP monomer. (**D**) Superimposition of monomeric $NP_{core}$ (*Olal and Daumke, 2016*) (in gray) and NP from the present structure (coloured as in *Figure 1A,B*). The $Ct_{arm}$ rotation is highlighted and newly built elements are labelled, shown as non-transparent and surrounded by dotted lines. A close up view of the $Ct_{arm}$/ $Nt_{arm}$ interaction is shown.

DOI: https://doi.org/10.7554/eLife.43075.003

The following source data and figure supplements are available for figure 1:

**Source data 1.** Cryo-EM data collection, refinement and validation statistics.
DOI: https://doi.org/10.7554/eLife.43075.007
**Figure supplement 1.** Purification, EM analysis and symmetry determination of HTNV-NC.
DOI: https://doi.org/10.7554/eLife.43075.004
**Figure supplement 2.** Resolution of HTNV-NC structure.
DOI: https://doi.org/10.7554/eLife.43075.005
**Figure supplement 3.** Specific antigenic sites are localised in variable regions of the NC surface.
DOI: https://doi.org/10.7554/eLife.43075.006

RMSD of 0.808 Å over 205 atoms (*Figure 1D*). N-terminal and C-terminal arms ($Nt_{arm}$, $Ct_{arm}$) are connected to each extremity of $NP_{core}$ by flexible hinges (*Figure 1B*, *Video 1*). Superimposition of HTNV-NC with the truncated monomeric structure (*Olal and Daumke, 2016*) shows that the $Ct_{arm}$ undergoes a 130.4° rotation upon multimerisation (*Figure 1D*). As a result, Nt- and $Ct_{arm}$ of the same subunit embrace each other: residues 97–98 of $Nt_{arm}$ contact residues 423–424 of $Ct_{arm}$, revealing a unique intra-arms interaction specific to HTNV-NC (*Figure 1D*, *Video 1*).

Recombinant HTNV-NC are rigid and remain stable in a large range of salt, pH and temperature conditions (*Figure 2—figure supplement 1C*). This can be explained by the multiplicity of

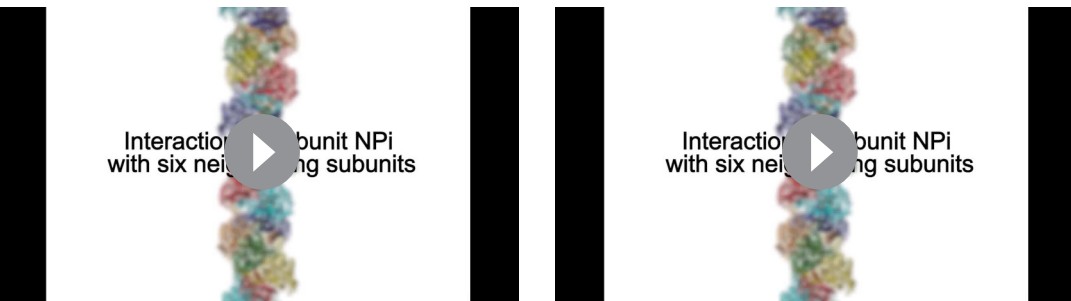

**Video 1.** Overview of HTNV-NC structural organisation. The helical arrangement of HTNV-NC is coloured as in *Figure 1A* upper part. HTNV-NP domain organisation is coloured as in *Figure 1B*. $Ct_{arm}$/$Nt_{arm}$ interaction is highlighted.

DOI: https://doi.org/10.7554/eLife.43075.008

**Video 2.** Interaction of subunit $NP_i$ with six neighbouring subunits. Subunit $NP_i$ (in green) interacts: (i) with subunit $NP_{i-1}$ (in yellow) via its $Nt_{arm}$, (ii) with subunit $NP_{i+1}$ (in orange) via its $Ct_{arm}$, with subunit $NP_{i+2}$ (in dark blue) and subunit $NP_{i+3}$ (in cyan) via its β-hairpin tip. Due to helical symmetry, $NP_{i-3}$ β-hairpin tip (in light cyan) also interacts with $NP_i$ (same interaction as $NP_i$-$NP_{i+3}$). Consistently, $NP_{i-2}$ β-hairpin tip (in red) also interacts with $NP_i$ (same interaction as $NP_i$-$NP_{i+2}$).

DOI: https://doi.org/10.7554/eLife.43075.011

interactions between protomers, each NP interacting with six other subunits (*Figure 2A*, *Video 2*). Successive subunit interactions rely on exchange of their $Nt_{arm}$, and $Ct_{arm}$ that make intimate contacts with the core domain of neighbouring protomers (*Figure 2A,B,C*), resulting in a buried area of 2704 Å$^2$ at each NP-NP interface. The $NP_i$ $Nt_{arm}$ forms an elongated structure that binds to residues 155–160, 177–181, 189–192 and 136–140 of the $NP_{i-1}$ subunit (*Figure 2B*, *Video 2*). Since constructs lacking the $Nt_{arm}$ remain monomeric (*Olal and Daumke, 2016*), the identified contacts appear to be essential for multimerisation. The $NP_i$ amphipatic $Ct_{arm}$ binds in a hydrophobic pocket of the protomer $NP_{i+1}$ comprising residues 334–346 and 378–394 (*Figure 2C*, *Video 2*). This agrees with the results of double hybrid experiments (*Kaukinen et al., 2004*; *Yoshimatsu et al., 2003*) which suggested that interaction of C-terminal helices of neighbouring protomers are critical to oligomerisation. Another key actor of multimerisation is the β-hairpin$_{152-181}$ that protrudes towards the exterior of the NP-core (*Figure 1B*). The two β-strands of the hairpin (residues 155–160 and 177–181) interact with the $Nt_{arm}$ residues 101–103 from the following subunit to form a 3-stranded β-sheet (*Figure 2A,B*, *Video 2*). As shown by pull-down assays of mutants L102A, V104A, this structure has a main contribution in NC stabilisation (*Guo et al., 2016*). In addition, the β-hairpin$_{152-181}$ tip (residues 162–175) acts as a clamp and seals the C-terminus-mediated contacts between $NP_{i+2}$ (residues 409 to 419) and $NP_{i+3}$ (residues 385–390), thereby buttressing and rigidifying the metastable helical form of the HTNV-NC (*Figure 2A,D*, *Video 2*).

HTNV recombinant NP was expressed in the absence of viral RNA, but after purification the optical density (OD) at 260–280 nm was measured to be around 1.0, which strongly suggests the binding of insect cell RNAs during expression. Consistently, three nucleotides with a partial occupancy can be visualised for each NP (*Figure 3A*). Conserved residues R197, R314, R368, R146 and K153 interact via salt bridges with RNA phosphates, in line with the versatility of RNA sequence to be incorporated by NP. In addition, F361 stabilises a stacking interaction network between the nucleotide bases. The nucleotides bind in a continuous positively charged groove oriented towards the interior of the NC. Density corresponding to additional nucleotides is visible in this groove, but present at low occupancy, suggesting the lower affinity of NC-RNA binding in these regions. Interestingly, the observed nucleotides occupancy is in accordance with electromobility shift assays of Sin Nombre NP mutants (*Guo et al., 2016*): indeed, in this work, residues directly interacting with the three visible RNA nucleotides were identified as the ones displaying the highest affinity with RNA. Altogether, these results reveal that HTNV-NC structure is compatible with binding of long viral RNA, thereby reinforcing the biological relevance of this structure (*Figure 3B,C*).

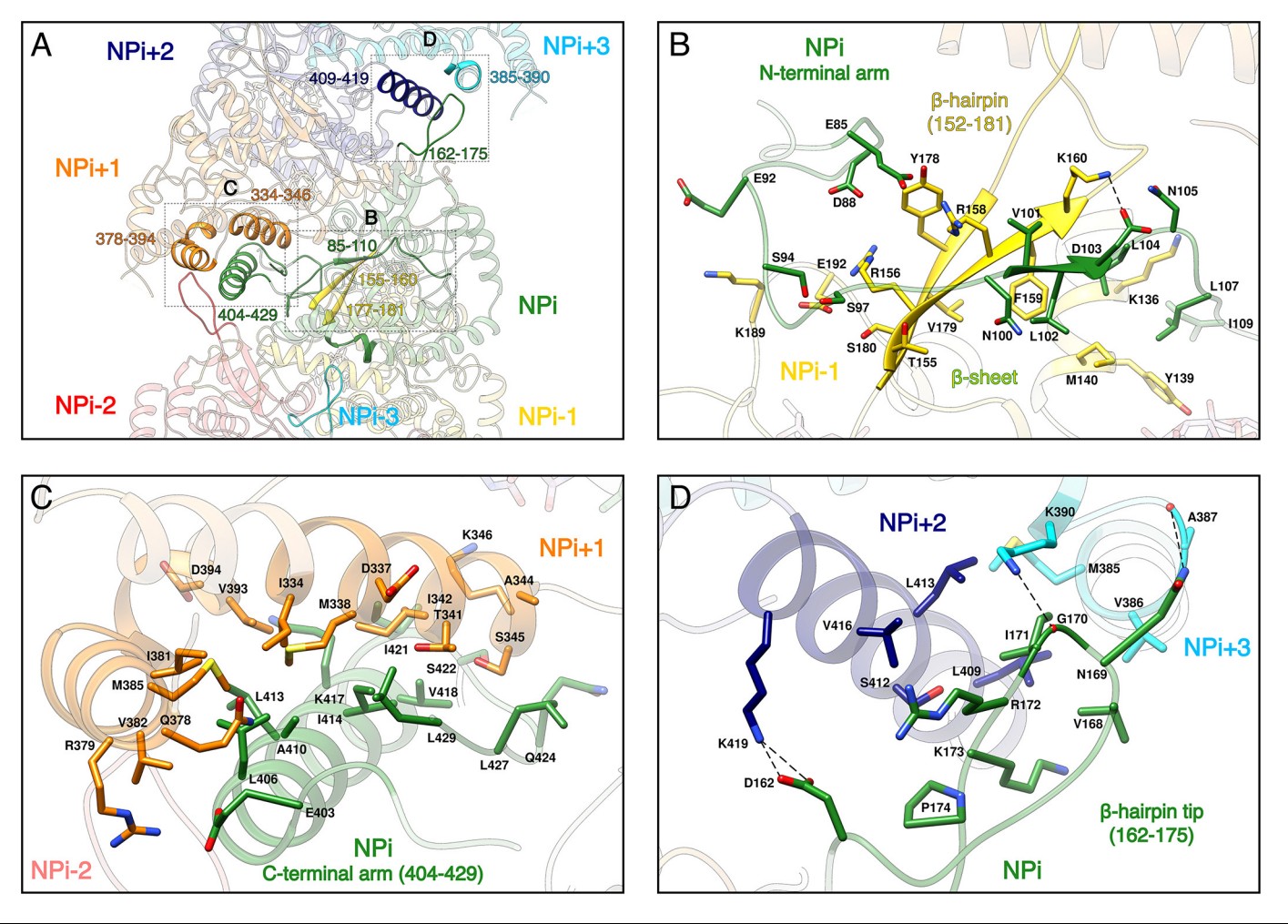

**Figure 2.** NP-NP interactions. (A) General view. The filament orientation corresponds to RNA direction from 5' to 3'. Each NP protomer is coloured differently. Interacting regions of NP$_i$ with subunits ranging from. NP$_{i-3}$ to NP$_{i+3}$ are shown as non-transparent. Positions of *Figure 2B,C,D* close-up views are indicated with dotted lines. (B) NP$_i$ Nt$_{arm}$ binding site in NP$_{i-1}$. Residues 101–103 from NP$_i$ Nt$_{arm}$ form a β-sheet with β-strands 155–160 and 177–181 from NP$_{i-1}$. Hydrogen bonds are shown as black dotted lines. (C) NP$_i$ Ct$_{arm}$ binding site in NP$_{i+1}$. (D) Binding of NP$_i$ β-hairpin tip (residues 162–175) on NP$_{i+2}$ (residues 409–419) and NP$_{i+3}$ (residues 385–390). NP$_i$ I171 is plugged into a hydrophobic pocket formed by L409 and L413 of the NP$_{i+2}$ C-terminal helix and M385 and K390 sidechains of the NP$_{i+3}$ C-terminal lobe. Hydrogen bonds between the residue pairs N169-A387 and G170-K390 from NP$_i$ and NP$_{i+3}$ respectively, further stabilise this interaction contributing to the rigidification of HTNV-NC.

DOI: https://doi.org/10.7554/eLife.43075.009

The following figure supplement is available for figure 2:

**Figure supplement 1.** Limited proteolysis and stability of recombinant HTNV-NCs.

DOI: https://doi.org/10.7554/eLife.43075.010

## Discussion

Structural determination of a full-length RNA-bound helical NSV NC at 3.3 Å resolution is particularly valuable because the usual intrinsic flexibility of NSV full-length NCs prevents their high-resolution analysis. It fits together the pieces of the jigsaw accumulated over several years of biochemistry and structural analysis on Hantavirus NP. HTNV recombinant NC structure indeed displays 3.6 subunits per turn and is thus consistent with observations of HTNV-NP trimers made by several groups (*Alfadhli et al., 2001*; *Kaukinen et al., 2004*). The present structure is also compatible with the proposed model (*Kaukinen et al., 2001*) which suggested that NPs first trimerise around the viral RNA and then gradually assemble to form longer multimers. The role of Nt$_{arm}$ and Ct$_{arm}$ exchange between successive subunits, identified by double-hybrid and pull-down experiments (*Guo et al.,*

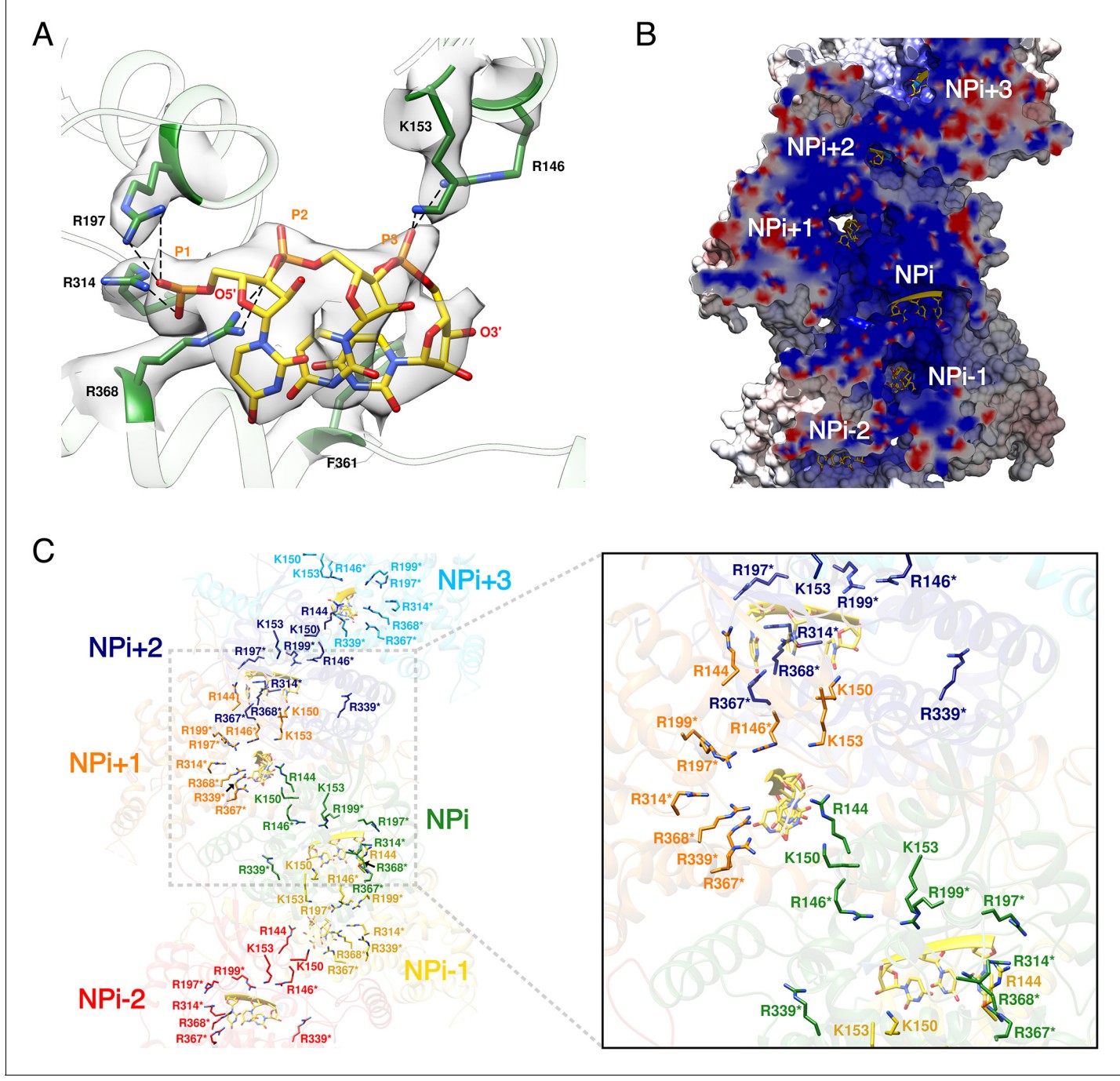

**Figure 3.** HTNV-NC RNA binding site. (**A**) RNA binding mode of 3 nucleotides. EM density of RNA and RNA-binding residues are displayed in transparent grey. Hydrogen bonds are shown as dotted lines. (**B**) Cut-away view of NC electrostatic potential showing the continuous positively-charged groove. RNA nucleotides are displayed. (**C**) RNA-binding residues are shown as sticks and coloured per subunit. RNA-binding residues defined in *Guo et al. (2016)*, namely R146*, R197*, R199*, R314*, R339*, R367*, R368* are labelled with stars, while RNA-binding residues identified in the present structure, namely R144, K150, K153 are shown without stars.

DOI: https://doi.org/10.7554/eLife.43075.012

The following figure supplement is available for figure 3:

**Figure supplement 1.** Comparison of LACV and HTNV-NP/NC, RNA-binding model.

DOI: https://doi.org/10.7554/eLife.43075.013

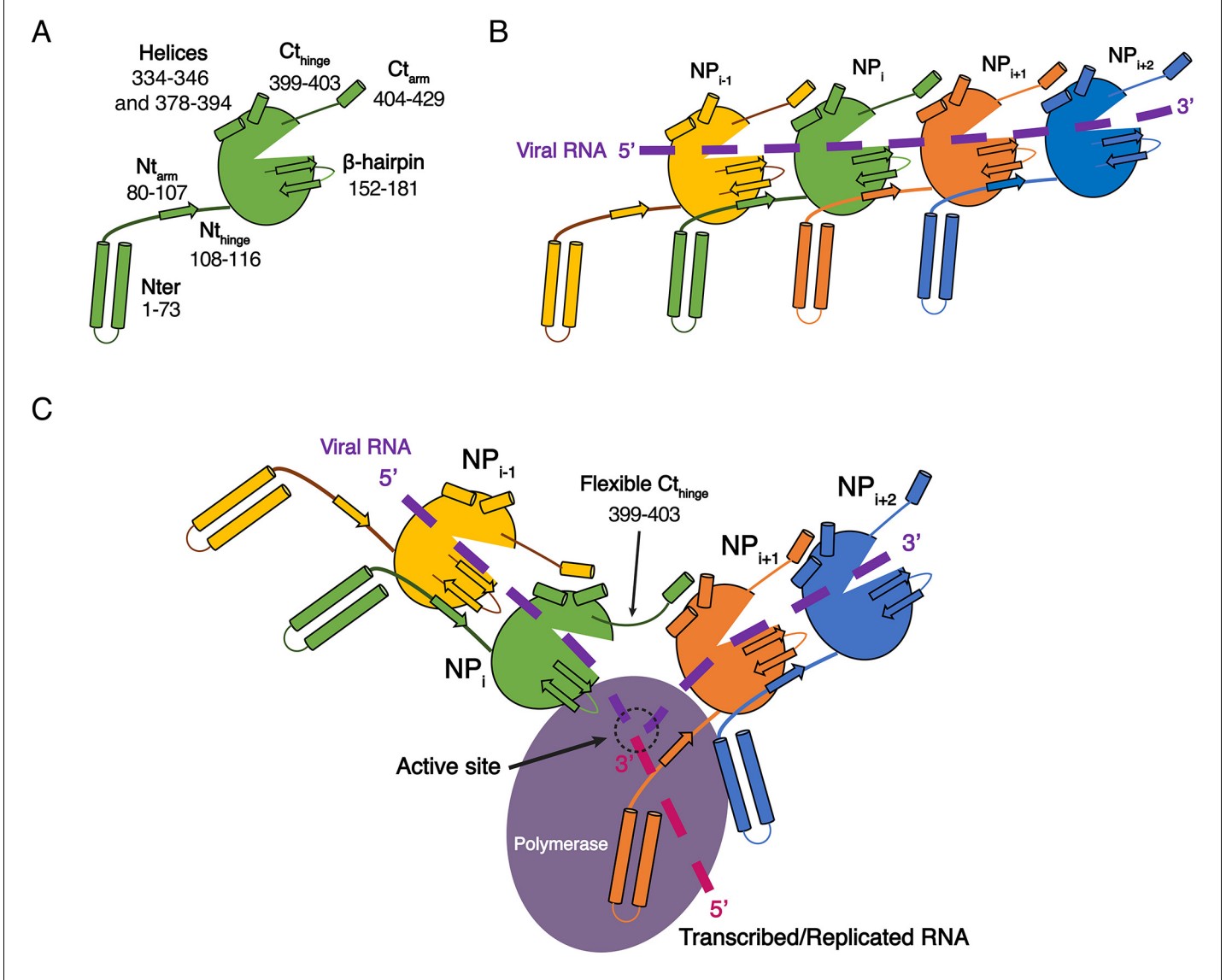

**Figure 4.** Model of HTNV replication and transcription. (**A**) Schematics representation of HTNV-NP. Major secondary structures involved in interprotomer interactions are represented as arrows for β-strand and cylinders for α-helices. RNA binding cavity is represented as a clipped part from the NP$_{core}$ region (green oval). (**B**) Schematic representation of RNA binding and NP-NP interactions. RNA is shown as a dotted purple line. Main NP-NP interactions between adjacent subunits are indicated. For clarity, NP$_i$ interactions with NP$_{i-2}$, NP$_{i-3}$, NP$_{i+2}$ and NP$_{i+3}$ are absent from the schematic representation. (**C**) Replication working hypothesis model which is inspired from *Gerlach et al. (2015)*. The polymerase is shown in purple and the newly transcribed/replicated RNA is indicated as a pink dotted line. The model proposes that the polymerase binds to the flexible N-terminal$_{1-73}$ region during replication/transcription in order to move along the NC. This would be reminiscent to P and L binding to flexible sNSV C-terminal region. Binding of the polymerase to the N-terminal$_{1-73}$ region could destabilise the adjacent Nt$_{arm}$ binding. This localised disruption of NP-NP interaction would create a local opening of the NC enabling transient access of the polymerase to few RNA nucleotides for replication/transcription. Such an opening could be possible without disturbing the whole NC as NP to NP contacts are driven not only by the Nt$_{arm}$ but also by the Ct$_{arm}$. The Ct$_{arm}$ interaction is likely to remain intact even with a local opening of the NC as the C-terminal hinge allows the Ct$_{arm}$ to undergo large rotation.
DOI: https://doi.org/10.7554/eLife.43075.014

2016; *Kaukinen et al., 2004*; *Yoshimatsu et al., 2003*), is in line with HTNV recombinant NC structure. These observations, together with the visualisation of a continuous positively charged groove, strongly suggest that the present structure is biologically pertinent. HTNV recombinant NCs are thus likely to be similar to helical NCs observed within viral particles, although the fact that the latter are less straight implies that at least the β-hairpin$_{152-181}$ might change conformation in the viral context,

thereby enabling more flexibility (*Battisti et al., 2011*; *Huiskonen et al., 2010*). Other conformations of NCs are in addition likely to exist because the 5' and 3' end of HTNV viral RNA are known to bind to the viral polymerase, implying that the NC must somehow be circularised (*Garcin et al., 1995*). Accordingly, flexible pearl-necklaces are also visible within virions, in infected cells and in NC extracted from viral particles (*Battisti et al., 2011*; *Goldsmith et al., 1995*; *Guo et al., 2016*; *Huiskonen et al., 2010*). Combination of helical NCs observed here and flexible pearl-necklaces thus represent relevant genome-encapsidating conformations.

HTNV and LACV NP$_{core}$ share a common fold (*Guo et al., 2016*; *Olal and Daumke, 2016*) enabling their structural superimposition (*Figure 3—figure supplement 1A*) and comparison of their RNA binding mode. This reveals that the three nucleotides present in HTNV-NC adopt similar conformations as the nucleotides 5, 6 and 7 of LACV-NP (*Figure 3—figure supplement 1B*) (*Reguera et al., 2013*). Key residues involved in the binding of HTNV-NC nucleotides, namely R197, R367 and F361 are conserved in LACV-NP (R94, R183 and Y177). Superposition of RNA-bound LACV-NP monomers on HTNV-NC shows that LACV RNA fits reasonably well into the HTNV-NC positively charged groove (*Figure 3—figure supplement 1C*). However, the proposed HTNV-NC RNA path is slightly shorter, suggesting that each HTNV-NP can contain between 8 and 10 nucleotides (*Figure 3—figure supplement 1C*). Modelling of RNA binding in HTNV-NC based on RNA position in LACV-NP (see Materials and methods) indicates that RNA phosphates and riboses interact with residues R146, K150, K153, R197, R314, R339, R367 and R368, while bases are surrounded by residues 113–116, 143–151, 182–188 and 217–222 (*Figure 3—figure supplement 1D*).

While the structures of HTNV and LACV NPs share a common fold for their cores, they however differ in their N-terminal organisation, HTNV Nt$_{arm}$ being significantly longer and linked to the unresolved N-terminal$_{1-73}$ region (*Figure 3—figure supplement 1A*). The N-terminal$_{1-73}$ region may bring the polymerase in close contact to NP as suggested (*Cheng et al., 2014*), and thus play a role similar to the intrinsically disordered phosphoproteins and NP C-terminal regions in nsNSV (*Ivanov et al., 2011*). One may therefore hypothesise that during genome reading by the polymerase, the Nt$_{arm}$ of one HTNV-NP could detach from its adjacent subunit, thereby affecting the conformation of the next NP β-hairpin and removing the seal over the two following NPs. This would provoke a local disruption of the metastable NC and provide RNA access to the polymerase, enabling replication and transcription (*Figure 4*). This opens new routes for future experiments, such as cryo-EM analysis of *in vivo* reconstituted mini-ribonucleoprotein particles (RNP) or cryo-electron tomography of viral RNPs, that would decipher the exact mode of interaction between NPs, polymerase and RNA during RNA replication/transcription. The rigidity of HTNV-NC and the absence of disordered phosphoprotein in HTNV should facilitate this study compared to the more complex NP/phosphoprotein/polymerase complex of nsNSV, and thus represents a unique opportunity to unravel significant properties of NSV replication.

In addition to these fundamental aspects, the HTNV-NC structure reveals the position of specific antigenic sites on variable regions of the NC surface (*Tischler et al., 2008*) thus providing a structure-based rationale for diagnosis (*Figure 1—figure supplement 3*). It may also stimulate the design of antivirals, because it defines key regions involved in NP oligomerisation and RNA encapsidation.

## Materials and methods

**Key resources table**

| Reagent type (species) or resource | Designation | Source or reference | Identifiers | Additional information |
|---|---|---|---|---|
| recombinant DNA reagent | *HTNV-NP* | Geneart | | Synthetic gene |
| Peptide, recombinant protein | HTNV-NP | This article | UniProtKB-P05133 | HTNV-NP was obtained by expression in insect cells of the synthetic gene *HTNV-NP* mentioned above |
| Software, algorithm | Motioncor2 | doi: 10.1038/nmeth.2472 | | http://msg.ucsf.edu/em/software/motioncor2.html |

*Continued on next page*

*Continued*

| Reagent type (species) or resource | Designation | Source or reference | Identifiers | Additional information |
|---|---|---|---|---|
| Software, algorithm | Gctf | doi: 10.1016/j.jsb.2015.11.003 | RRID:SCR_016500 | https://www.mrc-lmb.cam.ac.uk/kzhang/ |
| Software, algorithm | Relion2.1 and Relion3 | doi: 10.1016/j.jsb.2012.09.006 and doi: 10.1101/421123 | RRID:SCR_016274 | https://www2.mrc-lmb.cam.ac.uk/relion/index.php?title=Main_Page |
| Software, algorithm | Coot | doi: 10.1107/S0907444910007493 | RRID:SCR_014222 | https://www2.mrc-lmb.cam.ac.uk/personal/pemsley/coot/ |
| Software, algorithm | RCrane | doi: 10.1107/S0907444912018549 | | https://pylelab.org/software/rcrane-readme |
| Software, algorithm | PHENIX | doi: 10.1107/S0907444909052925 | RRID:SCR_014224 | https://www.phenix-online.org/ |
| Software, algorithm | Chimera | doi: 10.1002/jcc.20084 | RRID:SCR_004097 | https://www.cgl.ucsf.edu/chimera/ |
| Software, algorithm | Haddock | doi: 10.1038/nprot.2010.32 and 10.1016/j.jmb.2015.09.01 | | https://haddock.science.uu.nl/ |

## Cloning, Expression and Purification

Sequence-optimised synthetic DNA encoding a N-terminal his-tag, a TEV protease recognition site and the HTNV-NP (NCBI accession code NC_005218) was synthetised (Geneart) and cloned into a pFastBac1 vector between NdeI and NotI restriction sites.

The HTNV-NP-expressing baculovirus was generated via the standard Bac-to-Bac method (Invitrogen). For large scale expression, Sf21 cells at $0.5 \times 10^6$ cells/mL concentration were infected by adding 0.1% of virus. Expression was stopped 72 hr after the day of proliferation arrest.

The cells were disrupted by sonication 3 min (10 s ON, 20 s OFF, 40% amplitude) on ice in lysis buffer (20 mM Tris-HCl pH 8, 300 mM NaCl, 10 mM MgCl$_2$, 2 mM β-mercaptoethanol and 20 mM Imidazole) with EDTA free protease inhibitor complex (Roche). After lysate centrifugation at 20,000 rpm during 45 min at 4°C, protein from the soluble fraction was loaded on a nickel column, washed with 10 volumes of lysis buffer, 10 volumes lysis buffer supplemented with 50 mM Imidazole and eluted with 5 volumes of lysis buffer supplemented with 500 mM Imidazole. The eluted protein was cleaved with TEV protease in a 20:1 w/w ratio overnight at 4°C in dialysis against lysis buffer resulting in an almost complete cleavage. A second nickel column step was performed to remove unwanted material. The resulting protein was concentrated by ultracentrifugation using Optima XE SW55Ti rotor (Beckman Coulter) and 0.8 mL Ultra Clear tube during 2 hr at 45,000 rpm, 4°C. Concentrated HTNV-NC present in the 20 µl bottom fraction of each tube was gently resuspended. The 260/280 nm absorbance ratio was measured to be around one at the end of the purification indicating presence of nucleic acid.

Three biologically independent batches of large-scale expression and fifteen biologically independent batches of purifications were performed and gave reproducible results.

## Limited proteolysis by trypsin and N-terminal sequencing

Limited proteolysis of HTNV-NC was performed at 20°C in lysis buffer with a 3:2 w/w ratio of HTNV-NP/trypsin. Proteolysis was stopped by the addition of denaturing SDS-PAGE loading dye and incubation at 95°C for 5 min. Digestion was visualised in 10% acrylamide SDS-Page gels. For N-terminal sequencing, proteins were transferred on PVDF membrane previously incubated in 10 mM CAPS pH 11, 10% methanol buffer. PVDF membrane was stained using 0.1% Coomassie Brilliant Blue R-250, 40% methanol, 1% acetic acid buffer until bands were visible, washed using 50% methanol and dried. Visible bands were cut and subjected to N-terminal sequencing. Amino acid sequence determination based on Edman degradation was performed using an Applied Biosystems gas-phase sequencer model 492 (s/n: 9510287J). Phenylthiohydantoin amino acid derivatives generated at each sequence cycle were identified and quantitated on-line with an Applied Biosystems Model

140C HPLC system using the data analysis system for protein sequencing from Applied Biosystems (software Procise PC v2.1).

## Electron microscopy

For negative stain EM grid preparation, 4 µl of sample was applied between mica and carbon layer and stained using sodium silicotungstate (SST) 2%. After removing the mica part, the grid was deposited on the carbon layer and dried at room temperature. Micrographs were collected at 2.5 µm defocus using a FEI Tecnai F20 operated at 200 kV on 4 k*4 k CETA FEI CCD camera.

For cryo-EM grid preparation, Quantifoil grids 400 mesh 2/1 were glow-discharged at 30 mA for 1 min. 3 µl of sample were applied on the resulting glow-discharged grids and excess solution was blotted during 2.5 s force seven with a Vitrobot Mark IV (FEI) and the grid frozen in liquid ethane. Two biologically independent datasets, arising from different expression and purification batches, were collected on two high-end cryo-electron microscopes and gave consistent structures. The first cryo-EM dataset was collected on a FEI Polara F30 microscope operated at 300 kV equipped with a K2 summit GATAN direct electron detector and resulted in a 3.8 Å resolution structure. The second dataset, that gave rise to the 3.3 Å resolution structure presented in the present article, was collected on a FEI Titan Krios operated at 300 kV equipped with a Gatan Bioquantum LS/967 energy filter coupled to a Gatan K2 direct electron detector camera. For this second dataset, automated data collection was performed using EPU FEI software. Zero-loss micrographs were recorded at a 46,860x magnification giving pixel size of 1.067 Å with defocus ranging from 0.8 to 3.5 µm. In total, 4328 movies with 28 frames per movie were collected with a total dose of 40 e$^-$/Å$^2$.

## Image processing

Micrographs were initially selected based on visual quality inspection. Movie drift correction was performed in Motioncor2 (*Li et al., 2013*) excluding the two first frames. CTF parameters were determined in Gctf (*Zhang, 2016*) (RRID:SCR_016500). All subsequent processing steps were performed in Relion2.1 and Relion3 software (*Scheres, 2012*; *Zivanov et al., 2018*) (RRID:SCR_016274). Straight HTNV-NC were manually picked and computationally cut with an inter-box distance of 38 Å along the helical axis into overlapping boxes of 400*400 pixels, resulting in 168,709 extracted segments. 2D classification was used to eliminate bad quality filaments. The best 2D classes were aligned and padded in a square of 1200*1200 pixels $^2$. The individual 2D Power Spectra (PS) of each best class were averaged and used for a first estimation of the helical parameters using Fourier Bessel indexing (*Cochran et al., 1952*). An estimation of the repeat c was first inferred from the layer lines regularly spaced at multiples of ~335.6 Å$^{-1}$ (*Figure 1—figure supplement 1E*). Several strong meridional layer lines regularly spaced at multiples of 18.64 Å$^{-1}$ indicated the axial rise p between subunits. According to both c estimation and p, pitch P was inferred to be around 67.12 Å$^{-1}$ as a strong layer line with a first intensity near the meridian can be seen at l = 5. Therefore, the structure repeats after u = 18 subunits (18 (u) = 335.6 (c)/18.64 (p)), in t = 5 turns, resulting in a number of units per turn u/t of ~3.6.

3D helical reconstruction was performed using the 111,248 selected segments from the best 2D classes. A 130 Å diameter featureless cylinder was used as an initial model. Helical symmetry search was performed between −100 ± 4 ° for the helical twist (360°/u/t) and 18.64 ± 0.5 Å for the helical rise. Helical twist and rise after refinement were respectively −99.97° and 18.96 Å.

For the final 3D helical reconstruction, poorly aligned segments with an angular tilt >±20 ° were discarded resulting in 105,665 segments. Local CTF-determination was calculated for each segment using Gctf (*Zhang, 2016*) and finally CTF-refinement was performed using Relion 3.0[18]. The 10 last frames of each motion corrected micrographs were removed resulting in a total dose of ~22 e-/Å$^2$. The latest 3D map filtered at 15 Å resolution was used as initial model and helical symmetry search further refined during a last refinement. The resulting final reconstruction displays refined helical parameters of −99.95° for the twist and 18.87 Å for the rise. Post-processing, done using a B-factor of −103 Å$^2$, resulted in a 3.3 Å resolution reconstruction using the FSC 0.143 cutoff criteria (*Figure 1—figure supplement 2A*). Local resolution variations were estimated in Relion (*Figure 1—figure supplement 2B*).

Observation of nucleotides with partial occupancy suggests that only a fraction of HTNV-NC segments analysed contains RNA. We thus attempted to perform both 2D and 3D classifications in

order to separate RNA-bound and apo HTNV-NC segments using several strategies including usage of different masks, subtraction/absence of subtraction of protein density, symmetry release. However, due to the low molecular weight of RNA compared the protein (9 RNA nucleotides bound per HTNV-NP would correspond to 3 kDa of RNA versus 50 kDa of protein), these classifications did not succeed in separating the apo and RNA-bound states.

## Model building and refinement

The monomeric crystal structure of HTNV-NP comprising residues 113 to 429 (PDB code: 5FSG) was initially fitted into the EM density as two separate rigid bodies containing residues 113–398 and 399–429. Loops comprising residues 146–155 and 349–360, β-hairpin 160–172 and the N-terminal arm residues 80 to 112 - previously not present in the crystal structure - were manually built in Coot (*Emsley et al., 2010*). The three RNA nucleotides visible in the density map were manually built and adjusted using Coot (*Emsley et al., 2010*) (RRID:SCR_014222) and RCrane (*Keating and Pyle, 2012*). NP monomeric model was symmetrised according to the helical parameters to form a filament model of 7 subunits. Fiber model was then iteratively rebuilt and all-atom refined using stereochemical and NCS restraints within PHENIX (*Adams et al., 2010*) (RRID:SCR_014224).

## RNA modelling based on LACV NP RNA

In order to model HTNV RNA, two HTNV-NP were extracted from HTNV-NC, together with their respective three nucleotides (called 'HTNV-nts'). Five, six or seven LACV-nts were used to link the three HTNV-nts, giving rise to hybrid models containing between eight and ten nucleotides per NP. They were regularised in Coot, minimised in Chimera (*Pettersen et al., 2004*) (RRID:SCR_004097) and were then subjected to modelling in Haddock (*de Vries et al., 2010*; *van Zundert et al., 2016*). During the modelling procedure, the interaction between HTNV-NP and HTNV-nucleotides was maintained in the same conformation as in the EM map using unambiguous distance restraints. In addition, Haddock ambiguous interaction restraints were defined: residues involved in interactions (*Guo et al., 2016*) were considered as 'active', while all solvent accessible surface neighbours of active residues were defined as 'passive'. The initial starting orientations were not randomised. Rigid body minimisation, semi-flexible simulated annealing and flexible explicit solvent refinement was performed. For each starting RNA, the resulting models were very similar and gathered in one single cluster. The model with the lowest energy and the lowest Haddock score was thus considered as being the modelling result.

## Accession numbers

The atomic coordinates and the cryo-EM map have respectively been deposited in the PDB and EMDB under the accession codes 6I2N and EMD-0333.

## Acknowledgements

We thank Stephen Cusack for providing the synthetic gene and enabling JR and HM to initiate this project while being post-doctoral researchers in his lab. We thank Karine Huard, Angélique Fraudeau, Delphine Guilligay, Pascal Fender and Alice Aubert for technical advices on expression and purification; Jean-Pierre Andrieu for N-terminal sequencing; Maria Bacia and Michael Hons for help with cryo-EM data collection; Leandro Estrozi for advices on helical symmetry determination and discussion on image processing. This work used the platforms of the Grenoble Instruct-ERIC Center (ISBG: UMS 3518 CNRS-CEA-UGA-EMBL) with support from FRISBI (ANR-10-INSB-05–02) and GRAL (ANR-10-LABX-49–01) within the Grenoble Partnership for Structural Biology (PSB). The electron microscope facility is supported by the Rhône-Alpes Auvergne Region, the Fondation Recherche Medicale (FRM), the fonds FEDER and the GIS-Infrastrutures en Biologie Sante et Agronomie (IBISA). We acknowledge the European Synchrotron Radiation Facility for provision of microscope time on CM01. We thank all platform staff that enabled us to perform these analyses.

## Additional information

### Funding

| Funder | Grant reference number | Author |
| --- | --- | --- |
| Université Grenoble Alpes | G7H-IRS17H50 | Hélène Malet |

The funders had no role in study design, data collection and interpretation, or the decision to submit the work for publication.

### Author contributions

Benoît Arragain, Data curation, Validation, Investigation, Visualization, Methodology, Writing—original draft, Writing—review and editing; Juan Reguera, Ambroise Desfosses, Data curation, Investigation, Methodology, Writing—review and editing; Irina Gutsche, Resources, Data curation, Methodology, Writing—review and editing; Guy Schoehn, Conceptualization, Resources, Data curation, Supervision, Funding acquisition, Validation, Methodology, Project administration, Writing—review and editing; Hélène Malet, Conceptualization, Data curation, Supervision, Funding acquisition, Validation, Investigation, Visualization, Methodology, Writing—original draft, Project administration, Writing—review and editing

### Author ORCIDs

Benoît Arragain (iD) http://orcid.org/0000-0002-5593-4682

Juan Reguera (iD) http://orcid.org/0000-0003-4977-7948

Ambroise Desfosses (iD) http://orcid.org/0000-0002-6525-5042

Irina Gutsche (iD) http://orcid.org/0000-0002-1908-3921

Guy Schoehn (iD) http://orcid.org/0000-0002-1459-3201

Hélène Malet (iD) https://orcid.org/0000-0002-2834-7386

### Decision letter and Author response

Decision letter https://doi.org/10.7554/eLife.43075.023
Author response https://doi.org/10.7554/eLife.43075.024

## Additional files

### Supplementary files

• Transparent reporting form
DOI: https://doi.org/10.7554/eLife.43075.015

### Data availability

The cryo-EM has been deposited in EMDB under the accession code EMD-0333. The atomic coordinates have been deposited in PDB under the accession code 6I2N.

The following datasets were generated:

| Author(s) | Year | Dataset title | Dataset URL | Database and Identifier |
| --- | --- | --- | --- | --- |
| Arragain B, Reguera J, Desfosses A, Gutsche I | 2018 | Data from: High resolution cryo-EM structure of the helical RNA-bound Hantaan virus nucleocapsid reveals its assembly mechanisms | https://www.rcsb.org/structure/6I2N | Protein Data Bank, 6I2N |
| Arragain B, Reguera J, Desfosses A, Gutsche I | 2018 | Data from: High resolution cryo-EM structure of the helical RNA-bound Hantaan virus nucleocapsid reveals its assembly mechanisms | https://www.ebi.ac.uk/pdbe/entry/emdb/EMD-0333 | Electron Microscopy Data Bank, EMD-0333 |

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
