## [Decision Letter]

Thank you for submitting your article "High resolution cryo-EM structure of the helical RNA-bound Hantaan virus nucleocapsid reveals its assembly mechanisms" for consideration by *eLife*. Your article has been reviewed by three peer reviewers, including Edward H Egelman as the Reviewing Editor, and the evaluation has been overseen by John Kuriyan as the Senior Editor. The following individual involved in review of your submission has also agreed to reveal their identity: Matthias Wolf.

The reviewers have discussed the reviews with one another and the Reviewing Editor has drafted this decision to help you prepare a revised submission.

Summary:

The authors report an interesting structure of recombinant nucleocapsid of Hantaan virus HTNV, determined by cryo-EM. Compared to another *Bunyavirus*, La Crosse virus (LACV), there are similarities and differences. The essential core of N- and C-lobes is the same in both viruses, as well as other bunyaviruses. The main differences are the long N-terminal end and the C-terminal end. In HTNV, the ends assume quite a different structure compared to others. This leads to the suggestion by the authors that the terminal regions may be opened by the viral polymerase in order to read the sequence of the genomic RNA. Based on the structure and the flexibility of the nucleocapsid protein, this mechanism is very plausible. The manuscript is well and clearly written and focuses on essential points. Hantaan virus is pathologically highly relevant as it leads to illness with high fatality rate without any currently available vaccines.

Essential revisions:

1) Although their model of polymerase interaction is only deduced indirectly and not backed by their own experimental results, the authors clearly present it as a hypothesis. One additional experiment that would strengthen their claim may be cryo-tomography of HTNV-NP in the presence of viral polymerase, catching RdRp in the act near a kink of the HTNV-NP complex. But this is technically very challenging (probably requiring phase plate, movie processing and highest skills in cryo-ET), their construct may not be replication competent, and two months is a very short time. An easier one may be to ask for 2D class averages in presence of RdRp (ideally cryo, or at least negative stain), provided that the reaction can be carried out in vitro. This should be doable within 2 months and would add additional evidence to back up their interesting hypothesis (points 2 and 3).

2) The structure of three nucleotides per subunit was observed in the recombinant helical nucleocapsid, suggesting that the remaining nucleotides out of 8 to 10 per subunit are flexible. This raises questions about the inter-subunits observed in the recombinant helical nucleocapsid. What is reported here is one conformation of the nucleocapsid. As shown previously, the nucleocapsid in the viral particle and that extracted from the viral particle did not show a straight helical conformation. The authors need to discuss about the limitation of their current structure in the context of virus assembly.

---

## [Author Response]

Essential revisions:1) Although their model of polymerase interaction is only deduced indirectly and not backed by their own experimental results, the authors clearly present it as a hypothesis. One additional experiment that would strengthen their claim may be cryo-tomography of HTNV-NP in the presence of viral polymerase, catching RdRp in the act near a kink of the HTNV-NP complex. But this is technically very challenging (probably requiring phase plate, movie processing and highest skills in cryo-ET), their construct may not be replication competent, and two months is a very short time. An easier one may be to ask for 2D class averages in presence of RdRp (ideally cryo, or at least negative stain), provided that the reaction can be carried out in vitro. This should be doable within 2 months and would add additional evidence to back up their interesting hypothesis (points 2 and 3).

As requested, we attempted to obtain complexes of NC and RdRp (L) and image them by electron microscopy. Please note that this is very challenging as negative strand RNA virus RdRps are known to be difficult to handle. HTNV RdRp is a rather unstable protein, difficult to express/purify, and we are only in early phase of its characterization.

In order to maximize chances of success compared to a simple incubation of purified NC and purified L, we performed a co-expression of HTNV NP and L in insect cells. Co-purification however revealed that L and NC, although nicely purified, elute in different fractions (Author response image 1).

**Author response image 1. respfig1:** Purification and negative stain electron microscopy analysis of HTNV NP and L co-expression/co-purification. (**A**) SDS-PAGE gel of HTNV NP-L co-expression/co-purification that was done as follow: insect cells were co-infected with two baculoviruses respectively producing NP and L. Purification then consisted in Ni-NTA affinity and S200 gel filtration. (**B**) Fraction 2 of the SDS-Page gel contains HTNV-NC that are indistinguishable from HTNV-NC purified in the absence of L. (**C**) Fraction 6 of the SDS-Page gel contains homogeneous HTNV-L and very small HTNV-NC but no interaction is visible.

We want to point that this experiment does not invalidate our hypothesis. Indeed, interactions between NP and polymerase clearly exist *in vivo* and are necessary for replication and transcription. However, as spotted by the reviewers, it might not be possible to carry out this reaction *in vitro* or by co-expression/co-purification. Actually, all attempts of this kind performed by us on La Crosse virus failed (experiments performed by Juan Reguera, Piotr Gerlach and Hélène Malet during their postdoctoral/PhD research in Stephen Cusack group). Attempts of in vitro reconstitution of Influenza RNP that we are aware of were also unsuccessful (Stephen Cusack, personal communication).

We therefore think that it is very likely that only replication-competent active RNP would allow to observe NC-polymerase interaction. A replication-competent active RNP would contain a nucleocapsid bound to viral RNA, 3’ and 5’ extremities of viral RNA bound to the polymerase to initiate the reaction, and nucleotides to allow reaction to proceed. In order to obtain these RNPs and validate/invalidate our hypothesis, we think it would thus be necessary to either (i) reconstitute mini-RNP *in vivo* (as nicely done for influenza by Coloma et al., 2009, PLoS Pathogen) and image them by cryo-electron microscopy or (ii) perform cryo-electron tomography of viral RNP. As mentioned by the reviewers, these extremely interesting experiments are very challenging and cannot be carried out in two months. They would constitute another major advancement and would deserve a dedicated publication focusing on the NC-polymerase interaction.

We would therefore kindly suggest keeping the focus of our current article on HTNV nucleocapsid. The so-called model is only a hypothesis and we added a sentence to emphasize this point and make it clear to all readers: “One may therefore hypothesize that during…”. It is also clearly stated as a “Replication working hypothesis model”. Validation of this model is one of the main directions of our future research. We thus mention these long-term experiments as important perspectives:“This opens new routes for future experiments, such as cryo-EM analysis of in vivo reconstituted mini-ribonucleoprotein particles (RNP) or cryo-electron tomography of viral RNPs, which would decipher the exact mode of interaction between NPs, polymerase and RNA during RNA replication/transcription”.

2) The structure of three nucleotides per subunit was observed in the recombinant helical nucleocapsid, suggesting that the remaining nucleotides out of 8 to 10 per subunit are flexible. This raises questions about the inter-subunits observed in the recombinant helical nucleocapsid. What is reported here is one conformation of the nucleocapsid. As shown previously, the nucleocapsid in the viral particle and that extracted from the viral particle did not show a straight helical conformation. The authors need to discuss about the limitation of their current structure in the context of virus assembly.

We agree with the reviewers that what is reported is one conformation of the nucleocapsid. This important point was previously only described in the Supplementary Note 3. It is now fully integrated in the Discussion (see corresponding text copied in the next paragraph). As suggested by reviewers, we have also added the following points:

- We mention that helical NCs seen in the viral particles are not as straight as recombinant NCs (although bent recombinant NCs are also visible in a minority of EM micrographs, see Author response image 2).

**Author response image 2. respfig2:** Bent recombinant NCs seen in some EM micrographs. Bending localisations are shown with arrows.

- We discuss that potential differences in inter-subunit contacts may exist between recombinant NC and viral NC. We propose that theβ-hairpin_152-181_ tip, that is an essential element of recombinant NC rigidity, may be more flexible or change conformation in the viral context.

- We also mention that other conformations are present in virions, infected cells and RNPs extracted from viral particles. We are convinced that both helical and pearl-necklaces conformations are biologically relevant, and clearly state it in the article.

The corresponding text added in the article is the following:

“HTNV recombinant NCs are thus likely to be similar to helical NCs observed within viral particles, although the fact that the latter are less straight implies that at least the β-hairpin_152-181_ might change conformation in the viral context, thereby enabling more flexibility (Battisti et al., 2011; Huiskonen et al., 2010). […] Combination of helical NCs observed here and flexible pearl-necklaces thus represent relevant genome-encapsidating conformations.”